# Identification of Gene Responsible for Conferring Resistance against Race KN2 of *Podosphaera xanthii* in Melon

**DOI:** 10.3390/ijms25021134

**Published:** 2024-01-17

**Authors:** Sopheak Kheng, San-Ha Choe, Nihar Sahu, Jong-In Park, Hoy-Taek Kim

**Affiliations:** Department of Horticulture, Sunchon National University, Suncheon 57922, Republic of Korea; sopheakkheng097@gmail.com (S.K.); neosanha@naver.com (S.-H.C.); niharbio47@gmail.com (N.S.); jipark@scnu.ac.kr (J.-I.P.)

**Keywords:** melon, powdery mildew, R gene, LRR

## Abstract

Powdery mildew caused by *Podosphaera xanthii* is a serious fungal disease which causes severe damage to melon production. Unlike with chemical fungicides, managing this disease with resistance varieties is cost effective and ecofriendly. But, the occurrence of new races and a breakdown of the existing resistance genes poses a great threat. Therefore, this study aimed to identify the resistance locus responsible for conferring resistance against *P. xanthii* race KN2 in melon line IML107. A bi-parental F_2_ population was used in this study to uncover the resistance against race KN2. Genetic analysis revealed the resistance to be monogenic and controlled by a single dominant gene in IML107. Initial marker analysis revealed the position of the gene to be located on chromosome 2 where many of the resistance gene against *P. xanthii* have been previously reported. Availability of the whole genome of melon and its R gene analysis facilitated the identification of a F-box type Leucine Rich Repeats (LRR) to be accountable for the resistance against race KN2 in IML107. The molecular marker developed in this study can be used for marker assisted breeding programs.

## 1. Introduction

Melon (*Cucumis melo* L.) is a globally cultivated fruit crop, belonging to the *Cucurbitaceae* family, known for its sweet flavor, distinct aroma, and nutritional value in the market [1]. During the year 2019–2020, melon was cultivated with a production of 27.5 million tons throughout the world [2]. Over 25% of melon production is lost annually due to biotic stresses such as diseases and pests [3]. Most important among these are the bacterial fruit blotch, Anthracnose, Powdery mildew and Fusarium wilt [4].

Powdery mildew (PM), caused by the fungus *Golovinomyces cichoracearum* and *Podosphaera xanthii* (previously known as *Erysiphe cichoracearum* and *Sphaerotheca fuliginea*, respectively), is one of the most common and damaging foliar diseases in melons. Throughout the year, through natural infestation in fields and artificial infestation in greenhouse conditions [5,6] a numerous variety of vegetable crops and cereals are affected by this disease in various countries, with slight variations in their virulence [7,8,9,10,11,12]. According to Robinson and Decker-Walters [13], among the two fungi, *P. xanthii* is most widely reported on the melon.

*P. xanthii* and *G. cichoracearum* outbreaks occur in higher temperatures and humidity regions [7,14]. Typically, the symptoms start with the appearance of white powder-like patches on both the upper and lower side of the leaf’s surface and further spread to completely cover the entire plant [15]. Some symptoms, like chlorotic leaves and a reduced canopy, were commonly seen in many crops which leads to poor fruit quality and yield loss [16]. The intensity of the disease differs based on the melon cultivar, cultivation season, and geographical area which makes difficult to manage this disease [12].

The first occurrence of *P. xanthii* was reported in California in 1925. Since then, based on the reaction of *P. xanthii* isolates to melon differential lines, more than 28 physiological races were reported worldwide. Out of 28, three races—1, 2, and 3—were reported from the USA. Similarly, other races—0, 4, and 5—were reported in France and four races—1, N1 (race 6), N2 (race 7) and 5—were reported in Japan [12,17]. By comparing the interaction of 22 melon accessions against 28 races of *P. xanthii*, McCreight [18] identified eight variants of race 1 and six variants of race 2. In addition, Kim et al. [19] reported the existence of race N5, based on its distinctive reaction on 10 differential melon lines. Later, another two new races, named KN1 and KN2, were reported in South Korea, which were reported to be different in virulence from the previous reported races and also to each other, on the basis of differential lines [20]. In America and Brazil, the presence of races 1, 2 and 3 were reported [21,22]. In China, the predominance occurrence of races 1 and 2F were reported [23,24].

For many years, fungicides have been extensively used to control powdery mildew in melon, which is less effective and has led to the development of fungicide-resistant PM pathogens [25,26]. Therefore, the most effective way to manage this fungus is to develop resistant melon varieties [27,28,29]. Constant efforts of breeders has led to the development of more than 30 resistant germplasm worldwide against PM [18,22,30,31]. Extensive and wide cultivation of resistant varieties with a single gene has often resulted in the emergence of new virulent isolate or strain. Recently, breakdown of resistance in the melon varieties carrying single powdery mildew resistance gene was reported. Hence, it is necessary to develop melon varieties with durable resistance by pyramiding race-specific genes to overcome powdery mildew [12,22]. With the emergence of new races and break down of the resistance, it is necessary to identify the new sources of R genes, which will be a continuous process. 

Until now, several quantitative trait loci (QTL) conferring resistance against *P. xanthii* have been reported in melon. Most of them are located on chromosomes 2, 5, and 12 namely, *Pm1–6* [32,33,34], *Pm-R*, *W*, *X*, *Y* [28,29,35,36], *Pm-x1*, *x3*, *x5* [37], *Pm-Edisto47-1*, 2 [27], *Pm-2F* [38], *Pm-PxA*, *B* [39], *Pm-An* [24], *PmV.1*, *PmXII.1* [40], *BPm12.1* [41], and *pm-S* [42]. Identification of the putative candidate genes present in these QTL regions is important for breeding programs. The use of molecular markers that are closely linked to specific resistance genes can be a powerful tool for marker-assisted breeding [28].

Plants fight themselves against numerous diseases through diverse defense mechanisms. R-genes play a very vital role in these defense mechanisms by encoding putative receptors in response to the *Avr* genes released by the pathogen during the infestation [43]. Leucine-rich repeats (LRR) are one of the largest and important class of R genes in plants. The LRR domain can determine the R-gene’s specificity for various races of the same pathogen in different crops [44]. It was reported that LRR act as an effector-binding domain for the recognition of pathogens [45] and as a regulatory domain for signaling during the pathogen attack [46,47]. LRR were reported to confer disease resistance in many plant species against different diseases [48,49,50,51,52,53]. In the present study, we have identified and characterized the LRR genes present in the clusters of QTLs regions on the chromosomes 2, 5, and 12. We identified LRR as a candidate gene based on the physical position, structural diversity, and co-segregation against *P. xanthii* race KN2 in melon. We also developed a functional marker (LRR) for this gene, which may be further used in the marker assisted breeding.

## 2. Results

### 2.1. Pathogen Isolate and Determination of Physiological Race

Based on the report of Hong et al. [20], the KN2 isolate was characterized by using a set of eight differential melon lines. After 14 days of inoculation, scoring was noted for these eight lines. Out of eight differential lines, four melon lines, MR-1, PI124112, PMR5, and Edisto 47 were resistant to KN2, whereas the remaining four lines, SCNU1154, PMR 45, WMR 29 and PI414723 were susceptible (Figure 1 and Table 1). The results suggested that different patterns of pathogenicity were observed in the eight differential lines.

### 2.2. Inheritance of Resistance

Two breeding lines were screened against the race KN2. One line, namely IML107, showed resistance, whereas the remaining three breeding lines, namely IML197, were susceptible against the KN2 race (Figure 2). To understand the inheritance, the F_1_ derived from the cross between resistant IML107 and susceptible IML197 was found to be resistant against the KN2 race, whereas the phenotyping of 138 F_2_ were segregated into 100 resistant and 38 susceptible against KN2 (Appendix A). The inheritance of resistance was found to be complete dominant gene (100:38; 3:1). These results were further confirmed by the Chi square test (*p* = 0.68) (Table 2). 

### 2.3. Molecular Analysis of Resistance against Powdery Mildew Race KN2

In the present study, we have selected the four SNP markers on chromosomes 2, 5 and 12, reported to be associated with PM resistance and tested in the resistant and susceptible breeding lines. Among the four SNP markers tested, three marker Pm-2-HRM-1, Pm-5-HRM-1 and Pm-12-HRM-1 on chromosomes 2, 5 and 12 found to be polymorphic between the parents (Figure 3). These three markers were further tested in 100 R lines and 38 S lines of F_2_ populations. However, Pm-2-HRM-1 showed 94.28% segregation with the R and the S lines. The other two markers, Pm-5-HRM-1 and Pm-12-HRM-1, showed low percent segregation of 79.71% and 89.13%, respectively.

Based on the results of segregation analysis, we further explored the genes located on chromosome 2. All the 23 genes of LRRs and pathogenicity-related genes on chromosome 2 were performed for parental polymorphism along with its F_1_. Out of the 23 markers tested, marker 1, named MELO C2-1, corresponds F-box/LRR-repeat protein 15, marker 6 and marker 13, named MELO C2-6 and MELO C2-13, belong to the plant intracellular Ras-group-related LRR protein 6 and receptor protein kinase, putative, respectively, showing polymorphism between the parents and heterozygosity in the F_1_ tested (Table 3; Figure 4). These three markers were further tested with another mapping population developed between IML107 X IML197 with 39 F_2_ plants (Resistant-26; Susceptible-13) along with the parents. Out of three markers tested, F-box/LRR-repeat protein 15 (MELO C2-1) located at 562322 bases showed maximum segregation of 89.74% (0.00 cM) with the F_2_ phenotypes compared to the other two markers. Based on the linkage map, two gene-based markers, namely plant intracellular Ras-group-related LRR protein 6 (MELO C2-6) and receptor protein kinase, putative (MELO C2-13), showed 71.79% (33.13 cM) and 76.92% (65.16 cM) segregation, respectively (Figure 5) (Appendix A). Another marker (11) (Figure 4) showed polymorphism among the parents and was also able to differentiate the F_1_ but, when kept with the F_2_ plants, it did not show any segregation between them.

## 3. Discussion

Melon is a significant crop among the *Cucurbitaceae* family, cultivated globally. In South Korea, it holds economic importance and is widely grown to meet domestic demand as well as for its import purposes. It is a fruit of choice in many countries because of its various nutritional qualities [54,55,56]. However, the occurrence of powdery mildew, a fungal disease, is a serious threat to melon production by reducing yield and affecting the fruit quality. In melon, powdery mildew disease caused by *P. xanthii* is commonly reported when compared to the other fungus species. In South Korea, both race 1 and race 5 of powdery mildew have been documented [19,57]. This disease spreads widely in South Korea, and there are reports of the breakdown of existing R gene in various regions and existence of new races KN1 and KN2 [20]. The uses of molecular markers and identification of potential candidate genes linked to the powdery mildew resistance is a crucial requirement for the progression of molecular breeding [58]. It is important to develop the resistant varieties based on the physiological races of *P. xanthii* present on the particular region. Therefore, there is a continuous need for the identification of new resistance genes and the development of their molecular markers.

Even though a few dominant resistance genes were reported in melon against the powdery mildew, their genetic relationship is still not clear [59]. Most of the reported R genes and/or QTLs were mapped to the six melon linkage groups. The uses of molecular markers in the breeding have more advantages than the classical breeding. With the help of the molecular markers, we can identify the R gene/QTLs that linked to any phenotypic trait. These linked markers can also be used in the marker-assisted breeding program to pyramid more genes in single variety for durable resistance. The combination of dominant and recessive genes may play an essential role in conferring durable resistance in melon. 

There are many studies, reporting the presence of various races of *P. xanthii* causing powdery mildew and their resistant sources of melons in different countries [8,11,18,31,42,60,61,62,63,64,65,66,67,68]. Development of resistant varieties should be based on race specific and/or region specific factors. Up to this point, no reports of a resistance source were reported against the KN2 race of *Px*. This study is the first attempt made for the identification of resistant sources and understand the inheritance of resistance in melon against KN2 race. 

Our research group isolated and identified this race as a new strain of *P. xanthii* [20]. To evaluate race KN2 of powdery mildew, we used eight selected melon lines, and the results were the same as those reported by Hong et al. [20]. We have also screened four breeding lines against this KN2 race and identified the IML107 line to be resistant against the KN2 race. The inheritance of resistance of IML107 indicated that it is governed by the single dominant gene that controls the resistance against the KN2 race of *P. xanthii*. A similar inheritance study was reported in resistant line, Zimbabwean melon TGR-1551 resistance against 1, 2 and 5 of *P. xanthii.* Based on the genetic analysis, it is reported to carry two independent genes, one dominant and one recessive [69]. Similarly, the PMR 5 resistant line carries a single dominant gene against race 5 of *Px* [70]. Similarly, another dominant gene was reported from the PI 371795 cultivar and named the Pm-1 gene. Another PI 124112 accession was reported to be resistant against race 5 [21]. A single dominant gene identified in the PI 414723 accession confer resistance to race 1 and 2 of *P. xanthii* [35]. Another genotype WMR 29 was reported to carry a single dominant gene against the races 1, 2 and 3 of *P. xanthii* [36].

Previous studies have successfully mapped R-genes, and QTLs associated with *P. xanthii* resistance in melon on chromosomes 2, 5, and 12 [24,27,29,35,36,38,39,40,71,72]. Since most of the reported genes of *P. xanthii* were located on the chromosomes 2, 5 and 12 of melon genome. To further fine map the genes present in delimited regions of these chromosomes, the four SNP markers were identified and used for parental polymorphism and in F2 mapping population in our study. Our genotyping results showed an accuracy of 94.28%, with Pm-2-HRM-1 surpassing other markers previously employed. However, chromosome 2 may contain a potential gene associated with race KN2. Similarly, using the SNP markers putative candidate genes associated with race 5 specific were reported in *Cucumis melo* L. [73]. Two QTLs, Pm-R1-2, against races 1 and 2, and Pm-R5 against race 5 on LG V, which are reported to be linked to powdery mildew resistance in the TGR-1551 line [24,28,35,36]. Furthermore, using the SSR markers, the linkage analysis was performed and showed PmV.1 QTL against races 1, 2, and 3, and the PmXII.1 QTL against races 1, 2, and 5 on LG XII in the PI124112 line [29,40]. Similarly, around seventeen resistant genes have been located within the Pm12.1 resistant quantitative trait locus and in linkage group XII (chromosome XII), providing resistance against race 1 of powdery mildew in the melon line MR1 [39,41].

F-box/Leucine Rich Protein family proteins are one of the super protein families in plants, and several studies have demonstrated that they play diverse roles in various key plant development and physiological processes, including germination [74], regulation of hormone signaling transduction [75,76,77,78,79], plant response to stress conditions [80,81,82,83,84,85,86,87,88], and plant primary and secondary metabolism [89,90,91]. This gene is reported to be associated with the disease resistance in several plants [92,93].

In our study, one of the LRR gene showed 94% of segregation in the mapping population, suggesting that this could be the candidate gene involved in the resistance in IML107 against KN2 race of *Px*. Likewise, this class of gene were reported to be involved in defense against powdery mildew in wheat [94,95]. Through the whole genome-based approach, these NBS-LRR or LRR genes were found to be putative candidate genes for Pm1 gene in *Cannabis sativa* [96]. Similarly, in another study NBS-LRR was reported to be putative candidate gene in melon against powdery mildew [69]. Hence, all these findings suggest that the F-Box gene may be the putative candidate gene and the marker developed in this study will be used in the marker assisted breeding programs.

## 4. Materials and Methods

### 4.1. Plant Material and Screening against Pathogen

A set of eight differential lines are reported to differ in their reaction pattern against the powdery mildew races. These lines were screened to characterize against the race KN2 under glasshouse conditions at Sunchon National University (Figure 1 and Table 1).

Another set of four breeding lines were screened against KN2 race to identify the resistant and susceptible lines under glasshouse conditions at Sunchon National University (Figure 2).

### 4.2. Mapping Population

A mapping population was developed from a cross between IML197 (susceptible parent) and IML107 (resistant parent) in this study. From this cross around 15 F_1_ plants were obtained and checked for reaction against KN1 race. The positive F_1_ plant was selfed to produce the F_2_ populations. All the 138 F_2_ lines along with the parents were raised in nursery tray (50 holes) containing artificial soil mix under the controlled plant growth chamber with 25 ± 2 °C, 16 h day length, 80–120 μmol/m^2^/s light intensity and 65–75% relative humidity. All the 138 F_2_ population were subjected to screening against the KN2 race under glasshouse conditions.

### 4.3. Pathogen Isolate

The infected samples were collected from the melon fields in Jangheung, South Korea and maintained on the melon line SCNU1154 to obtain the pure culture. Before screening the mapping population, this pure culture was tested in the reported eight differential lines to check the reaction pattern. Later, this isolate was used for screening of P1, P2, F_1_ and 138 F_2_ plants under glasshouse conditions.

### 4.4. Inoculation and Disease Assessment

Plants were allowed to grow in the growth chamber until the third true leaf opened. At this stage the plants were inoculated with the culture by spraying. The inoculum was prepared by rinsing the spores from the heavily sporulating leaves in sterile water using a sterile paintbrush and then mixed with 0.02% Tween 20. The liquid suspension was filtered through a four-layered Mira-cloth (EMD Millipore Corporation, USA) to eliminate mycelia. The spore concentration of powdery mildew was adjusted with a haemocytometer and light microscope (Leica DM750, Leica, Switzerland) to measure a concentration of 5 × 10^5^ spores/mL by dilution with distilled water. This inoculum was sprayed on the plants using a hand sprayer. After two weeks of inoculation the disease was scored. The disease response on the melon lines was characterized visually as resistant, susceptible, and intermediate resistant based on the severity of the spread of fungal spores on the leaves of inoculated plants as estimated by the percentage of the infected area of leaf discs (0%–10%, 11%–30%, and 31%–100%, respectively) after two weeks of inoculation.

### 4.5. DNA Extraction

Genomic DNA was extracted from the 2-week-old fresh leaf samples of the two parental lines (15 from each parent), 15 F_1_ plants and 138 F_2_ plants using a DNeasy Plant Mini Kit (QIAGEN, Hilden, Germany) according to the manufacturer’s instructions. The concentration of extracted gDNA was checked using a NanoDrop Spectrophotometer ND-100 (NanoDrop Technologies, Wilmington, DE, USA) and stored at −20 °C for further use.

### 4.6. Genotyping with the Reported Molecular Markers

Up to this point, all the reported QTLs for PM resistance were located on the three chromosomes 02, 05 and 12 of melon genome. In this study, we have selected four linked markers from the three reported chromosomes. Based on the earlier reports of Zhang, Wang, and Li [24,38,41] the SNP sites near the reported locations were identified and HRM markers were designed. The SNP sites were used to design the probe. HybProbe was synthesized by Bioneer, Daejeon, Republic of Korea. A PCR was performed by using SYTO 9 green-fluorescent nucleic acid stain (Invitrogen, Waltham, MA, USA; Thermo Fisher Scientific, Waltham, MA, USA) to generate melting curves characteristic for the genotype corresponding to the probe. The reactions were carried out in a final volume of 10 μL containing the HRM PCR mix of 1 μL genomic DNA at 50 ng μL^−1^, 5 μL HS prime LP premix (GENETBIO, Daejeon, Republic of Korea), and 2.6 μL double-distilled H_2_O, 0.1 μL forward and 0.5 μL reverse primers (10 pmol), 0.5 μL probe (10 pmol), 0.3 μL SYTO 9 fluorescent dye. The PCR before the HRM was conducted with the following conditions: an initial preincubation at 95 °C for 5 min followed by 45 cycles of 95 °C for 20 s, annealing at 64 °C and 56 °C for 15 s under touchdown command, i.e., by decreasing temperature by 1 °C at each step and then extension at 72 °C for 15 s. Melt analysis was recorded by ramping the temperature to 95 °C for 60 s to 40 °C for 60 s and 97 °C for 1 s with continuous acquisition of fluorescence. HRM curve analysis was conducted using LightCycler^®®^ 96 SW1.1 software (Roche, Mannheim, Germany) at 75% discrimination for both delta Tm and curve shape with a 0.2 positive/negative threshold level.

### 4.7. Identification of Candidate Genes

Based on the earlier reports and our analysis on the reported linked markers, we targeted the chromosome 2 for Insilico analysis. Based on the earlier reports we have selected all the LRR, TIR-LRR, CC-LRR, NB-LRR and Pathogenesis related genes by using keyword search in the cucurbit database (http://cucurbitgenomics.org/; accessed on 2 September 2022). Around 23 genes were selected as putative candidate genes for further analysis. The primers were designed for these 23 genes using the primer3 web-based tool (https://primer3.ut.ee/; accessed on 14 September 2022). All these 23 primer pairs were used for parental polymorphism and their sequence details were given in (Appendix A). A linkage map was prepared using the linkage function of the QTL cartographer 2.0.

### 4.8. Statistical Analysis

Microsoft excel was used to calculate discrepancies between observed data and anticipated segregation ratios using a chi-square test for goodness-of-fit. The threshold for statistical significance was *p* < 0.05.

## Figures and Tables

**Figure 1 ijms-25-01134-f001:**
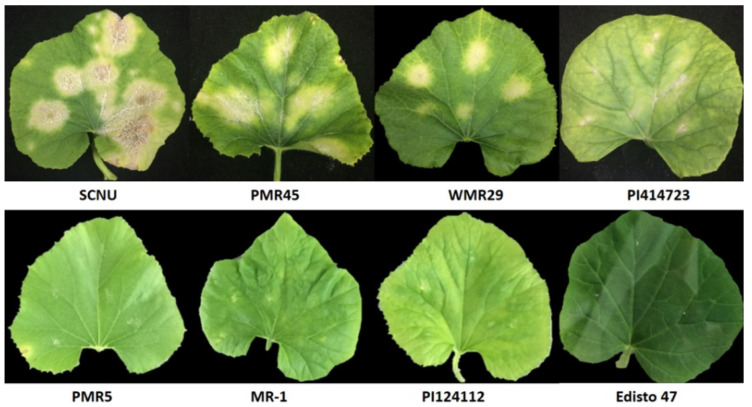
Disease assessment on eight differential cultivars against race KN2.

**Figure 2 ijms-25-01134-f002:**
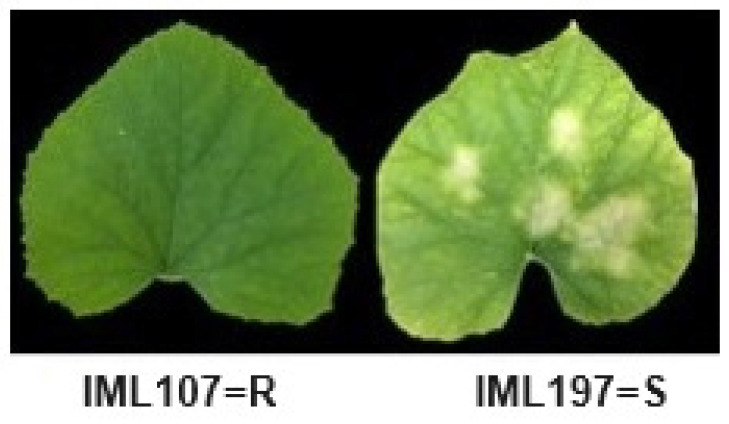
Disease reaction on resistant melon line IML107 and susceptible melon line IML197 against race KN2; R = Resistant; S = Susceptible.

**Figure 3 ijms-25-01134-f003:**
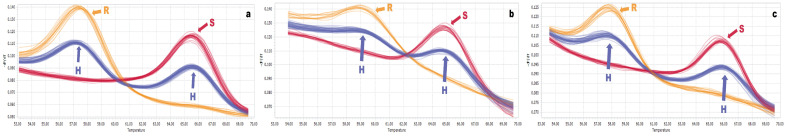
High Resolution Melting analysis with PM resistant markers (**a**) Pm-2-HRM-1 on Chr2, (**b**) Pm-5-HRM-1 on Chr5 and (**c**) Pm-12-HRM-1on chr12; Yellow curves represent R lines, red curves represent S lines and purple curves represents the heterozygotes which are marked by yellow, red and purple arrows, respectively.

**Figure 4 ijms-25-01134-f004:**
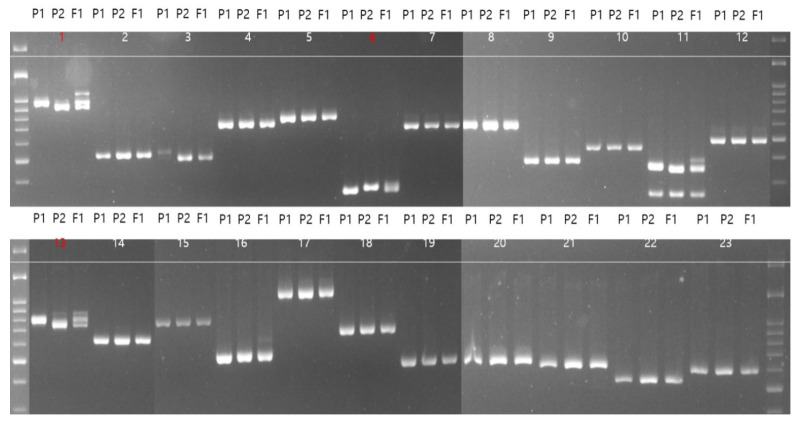
Gel picture showing the amplification pattern of 23 LRR primers on chromosome 2; P1—IML107 DNA, P2—IML197 DNA, F_1_—IML107 × IML197 DNA, numbers in red indicates the polymorphic marker.

**Figure 5 ijms-25-01134-f005:**
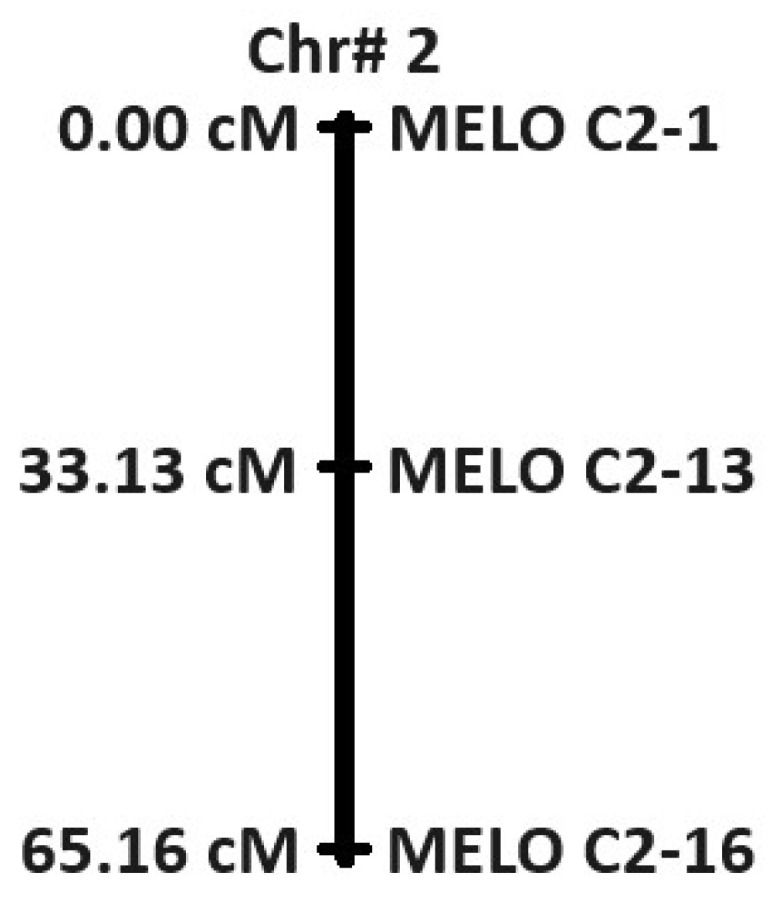
Linkage Map for three selected markers on chromosome 2 tested with 39 F_2_ plants.

**Table 1 ijms-25-01134-t001:** Characterization of race KN2 on eight differential cultivars.

S. No.	Melon Line	*Podosphaera xanthii*
		KN2
1.	SCNU1154	S
2.	PMR45	S
3.	WMR29	S
4.	PI414723	S
5.	PMR5	R
6.	MR-1	R
7.	PI124112	R
8.	Edisto 47	R

R = Resistant; S = Susceptible.

**Table 2 ijms-25-01134-t002:** Genetic inheritance of resistance against race KN2.

Materials	Plant Type	Resistant	Susceptible	Expected Ratio (R:S)	Chi-Square	*p* Value
IML107	P1	15	0	-	-	-
IML197	P2	0	15	-	-	-
IML107 × IML197	F_1_	15	0	-	--	
IML107 × IML197	F_2_	100	38	3:1	0.17	0.68

**Table 3 ijms-25-01134-t003:** Information of polymorphic LRR markers in the study.

No.	Marker Name	‘F’-Sequence	‘R’-Sequence	Product Size (bp)
1	MELO C2-1	GAAGGAGGTGAGGTTGGATAAG	TTTGGAACTGCTTGGGAAATG	997
2	MELO C2-6	ACTTGGTTGCCGTCCATTAT	GGTGTCTGGTAGCCTTGTTAG	360
3	MELO C2-13	GCGGAGGGATTGGCTTATT	CAGCATGCACCCGTATTCTA	805

## Data Availability

The data is contained within the article and Appendix A.

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
