# Peer review of "Identification of Gene Responsible for Conferring Resistance against Race KN2 of Podosphaera xanthii in Melon"

_ijms, 2024, doi:10.3390/ijms25021134_

Round 1

Reviewer 1 Report

Comments and Suggestions for Authors

The manuscript of Kheng et al., submitted to MDPI IJMS (ijms-2767539) reports the identification of a resistance gene in melon that causes resistance to Podosphaera xanthii, the causal agent of powdery mildew of cucurbitaceous plants.

The results obtained in the manuscript are important and worthy of publication in IJMS. However, the manuscript describing the results needs improvements.

In the Materials and Methods section some methods were not thoroughly described (see below).

A serious flaw of the authors was that they did not include supplementary material in the submission (or at least no supplementary material did get to me), so those could not be assessed.

The Discussion feels quite repetitive, and contains sections that should be moved to the Introduction. There is no real discussion of the results.

My questions and comments:

Line (L) 38: Does powdery mildew appear on the leaves’ lower side first?

L51-52: How much do P. xanthii isolates from Korea differ in their virulence?

L104 and elsewhere: Are the data and experiments on lines IML168 IML135 really necessary in the text? I feel they are not really adding much but only complicate understating. If they are needed, please be more explanatory in the text.

I think sections 2.2 and 2.3 are the same.

L133: The figure legends here should be more explanatory.

L137 and onwards: Again, please be more explanatory. Which marker corresponds to which gene? What about marker no. 11? This also looks polymorphic to me.

L147: Please describe how genetic distances are calculated.

Figure 3 is really not informative, in my opinion.

I would have been happy to read a little more about the newly discovered gene. You should check on the structure, sequence, its relatives, homologs, etc. What else is known about MELO C2-1?

L258: (Approximately) How long does it take for the plants to develop three true leaves?

L261: How old were P. xanthii colonies used for inoculum preparation?

L283: Please describe in detail how CAPS markers were converted to probes. How were primers designed?

L293: Please specify the touchdown settings.

L295: Please specify the recording of the HRM curve.

L311-316. I think you can remove these parts.

L321. Improvised…? 

Altogether, I suggest revisions to be done to create a higher-quality manuscriptt.

Comments on the Quality of English Language

Unfortunately, the English of the text is quite poor. There are tremendous amounts of grammar mistakes, incomplete sentences, typos and bad word choices. This is not only a minor issue but also really complicates understanding the text. I strongly suggest the authors ask a college who is native in English or use a language editing service to proofread the manuscript and improve the text.

Author Response

The manuscript of Kheng et al., submitted to MDPI IJMS (ijms-2767539) reports the identification of a resistance gene in melon that causes resistance to Podosphaera xanthii, the causal agent of powdery mildew of cucurbitaceous plants.

The results obtained in the manuscript are important and worthy of publication in IJMS. However, the manuscript describing the results needs improvements.

Response: Thank you for the comment, now the results has been improvised in the manuscript.

In the Materials and Methods section some methods were not thoroughly described (see below).

Response: Thank you for the comment, the methodology has now properly described.

A serious flaw of the authors was that they did not include supplementary material in the submission (or at least no supplementary material did get to me), so those could not be assessed.

Response: Thank you for the comment, the supplementary files has now been submitted along with the manuscript.

The Discussion feels quite repetitive, and contains sections that should be moved to the Introduction. There is no real discussion of the results.

Response: Thank you for the comment, now the discussion section has been improved in the manuscript.

My questions and comments:

Line (L) 38: Does powdery mildew appear on the leaves’ lower side first?

Response: Thank you for the comment, powdery mildew occurs on both side of the leaves as patches, which grows further. The sentence has been changed in the manuscript (page 1, line 38-40).

L51-52: How much do P. xanthii isolates from Korea differ in their virulence?

Response: Thank you for the comment, the two isolates from Korea also differ from each other on the basis of differential cultivar screening, which was reported in our previous paper Hong et al, 2018. According to that paper, out of eight differential lines used, race KN1 was virulent to 6 lines whereas race KN2 4 differential lines. So they have a different virulence pattern for both the races. The sentence is also modified accordingly (page 2, line 51-54)

L104 and elsewhere: Are the data and experiments on lines IML168 IML135 really necessary in the text? I feel they are not really adding much but only complicate understating. If they are needed, please be more explanatory in the text.

Response: Thank you for the comment, the data regarding IML 168 and IML 135 has been removed from the manuscript.

I think sections 2.2 and 2.3 are the same.

Response: Thank you for the comment, previous section 2.3 has been removed and section 2.4 has been changed to section 2.3.

L133: The figure legends here should be more explanatory.

Response: Thank you for the comment, the figure legend has been changed to be more explanatory (page 4, line 127-130)

L137 and onwards: Again, please be more explanatory. Which marker corresponds to which gene? What about marker no. 11? This also looks polymorphic to me.

Response: Thank you for the comment, the sentence has been rewritten to be more explanatory in the manuscript (page 4, line 133-138 and line 145-147).

L147: Please describe how genetic distances are calculated.

Figure 3 is really not informative, in my opinion.

Response: Thank you for the comment, Figure 3 and its legend is now more informative.

I would have been happy to read a little more about the newly discovered gene. You should check on the structure, sequence, its relatives, homologs, etc. What else is known about MELO C2-1?

Response: Thank you for the comment, information about MELO C2-1’s structure, sequence is now not known since we have not sequenced the gene. We are planning it in the future studies. Now only it is known that it belongs to the F-box/LRR-repeat protein. Once the gene is sequenced in IML107 and IML197 then only we can know the exact structure difference and other information.

L258: (Approximately) How long does it take for the plants to develop three true leaves?

L261: How old were P. xanthii colonies used for inoculum preparation?

L283: Please describe in detail how CAPS markers were converted to probes. How were primers designed?

Response: Thank you for the response, the sentence has been changed and modified (page 8, line 280-282)

L293: Please specify the touchdown settings.

Response: Thank you for the comment, the touchdown settings have been now specified in the manuscript (page 9, line 290-291).

L295: Please specify the recording of the HRM curve.

Response: Thank you for the comment, recording of the HRM curve is now specified in the manuscript (page 9, line 292-294).

L311-316. I think you can remove these parts.

Response: Thank you for the comment, these parts has been removed from the manuscript.

L321. Improvised…? 

Response: Thank you for your patient reading and observation. Now the manuscript has been improvised.

Altogether, I suggest revisions to be done to create a higher-quality manuscriptt.

Comments on the Quality of English Language

Unfortunately, the English of the text is quite poor. There are tremendous amounts of grammar mistakes, incomplete sentences, typos and bad word choices. This is not only a minor issue but also really complicates understanding the text. I strongly suggest the authors ask a college who is native in English or use a language editing service to proofread the manuscript and improve the text.

Response: Thank you for the comment, the overall English language of the manuscript has been improvised.

Reviewer 2 Report

Comments and Suggestions for Authors

In the keywords, we avoid repeating words from the title to enhance the text's detectability.

The introduction provides crucial information on the global significance of melons, listing the main diseases and their symptoms. I believe the introduction serves as a sufficient preamble.

Analyzing the work from a phytopathologist's perspective rather than that of a geneticist, I consider the method of melon inoculation to be correct. However, I am curious whether different plant lines were stored separately to prevent secondary infections. Was there any form of plant shielding after spraying, perhaps to regulate humidity?

Author Response

In the keywords, we avoid repeating words from the title to enhance the text's detectability.

 Response: Thank you for the comment, we think these key words are more appropriate for the manuscript.

The introduction provides crucial information on the global significance of melons, listing the main diseases and their symptoms. I believe the introduction serves as a sufficient preamble.

Analyzing the work from a phytopathologist's perspective rather than that of a geneticist, I consider the method of melon inoculation to be correct. However, I am curious whether different plant lines were stored separately to prevent secondary infections. Was there any form of plant shielding after spraying, perhaps to regulate humidity?

Response:Thank you for the comment, yes the lines were separately all the time and also the plants were shielded with plastic sheet for maintaining the humidity.
